# Giant Atrial Dilatation: Systematic Review of Reported Cases from the Last Decade and an Illustrative Case with Dysphagia and Severe Dysphonia

**DOI:** 10.3390/jcm14217832

**Published:** 2025-11-04

**Authors:** Caius Glad Streian, Iulia-Raluca Munteanu, Marinela-Adela Scuturoiu, Alina-Ramona Cozlac, Ana Lascu, Raluca-Elisabeta Staicu, Lucian-Silviu Falnita, Adrian Grigore Merce, Horea Bogdan Feier

**Affiliations:** 1Department VI Cardiology-Cardiovascular Surgery Clinic, “Victor Babes” University of Medicine and Pharmacy, 300041 Timisoara, Romania; streian.caius@umft.ro (C.G.S.); horea.feier@umft.ro (H.B.F.); 2Institute for Cardiovascular Diseases of Timisoara, Clinic of Cardiovascular Surgery, Gheorghe Adam Street, No. 13A, 300310 Timisoara, Romania; alina-ramona.cozlac@umft.ro (A.-R.C.); lascu.ana@umft.ro (A.L.); raluca.staicu@umft.ro (R.-E.S.); lfalnita@cardiologie.ro (L.-S.F.); adrian-grigore.merce@umft.ro (A.G.M.); 3Doctoral School Medicine-Pharmacy, “Victor Babes” University of Medicine and Pharmacy, 300041 Timisoara, Romania; 4Advanced Research Center, Institute for Cardiovascular Diseases of Timisoara, 300310 Timisoara, Romania; 5NEUROPSY-COG Center for Cognitive Research in Neuropsychiatric Pathology, Department of Neuroscience, “Victor Babes” University of Medicine and Pharmacy, 300041 Timisoara, Romania; 6Department III of Functional Sciences, Discipline of Pathophysiology, “Victor Babes” University of Medicine and Pharmacy, 300041 Timișoara, Romania; 7Centre for Translational Research and Systems Medicine, “Victor Babes” University of Medicine and Pharmacy, Eftimie Murgu Square No. 2, 300041 Timisoara, Romania

**Keywords:** giant left atrium, atrial fibrillation, dysphagia, atrial reduction surgery, reintervention

## Abstract

**Background/Objectives:** Giant atrial chambers are rare but clinically important conditions, most often linked to rheumatic mitral valve disease, though they may also occur in congenital or other acquired disorders. Despite their low prevalence, they entail major hemodynamic, arrhythmogenic, and extracardiac risks. This study aimed to review recent evidence on giant atrial pathology—including giant left atrium (GLA), giant right atrium (GRA), and atrial appendage aneurysms—and to illustrate its relevance through cases of symptomatic extracardiac compression. **Methods:** A PubMed search on 15 September 2025 using “giant atrium” and limited to human, free full-text studies from the last 10 years yielded 93 results. After screening, 21 reports describing 24 cases were analyzed and compared with institutional experience. **Results:** GLA is most often defined by an anteroposterior diameter ≥6.5 cm or ≥8 cm, while criteria for GRA and appendage aneurysms remain inconsistent. Reported complications include atrial fibrillation, thromboembolism, and compression of mediastinal structures, with presentations such as dysphagia or airway obstruction. While valve surgery alone may suffice, many authors recommend concomitant atrial reduction or aneurysm resection in symptomatic patients. **Conclusions:** Giant atrial pathology, though uncommon, carries significant cardiac and extracardiac implications. Management should be individualized, and awareness of atypical manifestations is critical for timely diagnosis and treatment.

## 1. Introduction

Extreme enlargement of the atrial chambers, most often described as giant left atrium (GLA), represents a rare but clinically significant condition. First reported in the nineteenth century, GLA has been classically associated with chronic rheumatic mitral valve disease, particularly in the presence of combined stenosis and regurgitation. Although definitions vary across studies, an anteroposterior diameter greater than 6.5 cm on echocardiography, or ≥8 cm in some series, is generally accepted as the diagnostic threshold. The reported incidence remains low, approximately 0.3–0.6% among patients with rheumatic heart disease [1,2,3].

While left atrium has been the primary focus of most reports, other forms of extreme atrial enlargement are increasingly recognized. Giant right atrium (GRA), though far less common, has been described in association with tricuspid valve disease, congenital malformations, or idiopathic dilatation [1,4]. Similarly, atrial appendage aneurysms—most frequently involving the left but occasionally the right appendage—constitute another rare manifestation with overlapping clinical risks [4,5].

The clinical importance of these entities derives not only from their rarity but also from their complications [2,6,7,8,9,10,11]. Severely enlarged atria create a favorable substrate for atrial fibrillation. This arrhythmia further promotes atrial remodeling and increases the risk of systemic or pulmonary thromboembolism. Massive atrial enlargement may also produce external compression of mediastinal structures. Such compression can lead to dysphagia, airway obstruction, or recurrent laryngeal nerve palsy. These features contribute to excess morbidity and mortality, including elevated rates of stroke and sudden cardiac death [7,12,13,14].

From a therapeutic perspective, consensus remains limited [6,13,15,16]. Some authors advocate for correction of the underlying valvular lesion alone, while others recommend concomitant atrial reduction or aneurysm resection, particularly in patients with compressive symptoms or recurrent embolic events. Recognition of the full spectrum of giant atrial pathology—encompassing GLA, GRA, and atrial appendage aneurysms—is therefore essential to guide timely diagnosis and individualized management.

## 2. Materials and Methods

The systematic review was prompted by the presentation of a patient with extreme left atrial dilatation and dysphagia at our institution. To better contextualize this clinical presentation and guide therapeutic decisions, we conducted a systematic literature search.

A structured query was performed in PubMed on 15 September 2025, using the following Boolean string: (“giant atrium” OR “giant left atrium” OR “giant right atrium” OR “atrial appendage aneurysm”) AND (“case report” OR “case series”). Filters were applied to include only human studies, free full-text articles, and publications from the last ten years (2015–2025). Only English-language articles were considered; non-English and subscription-based reports were excluded. No additional databases (e.g., Embase, Scopus) were searched, as PubMed provided adequate coverage for the targeted case-level literature. The search initially identified 93 records. Titles and abstracts were screened by two reviewers, and after full-text evaluation, 21 publications were retained, describing a total of 24 individual cases. A summary of the published cases is provided in Table A1 (Appendix B).

We focused on case reports and case series that provided original patient data. Studies were included if they reported clinical presentation, diagnostic findings, management, or outcomes. Reviews without primary cases, animal studies, abstracts without full text, and duplicates were excluded.

For each eligible case, we extracted information on demographics, presenting symptoms, comorbidities, diagnostic methods, atrial dimensions, arrhythmias, complications, treatment strategies, and patient outcomes. Data extraction was performed independently by two authors and cross-checked for accuracy.

Discrepancies were resolved by consensus discussion to ensure consistency and minimize subjective bias. Given the descriptive and heterogeneous nature of the included case reports, no formal risk-of-bias score was applied; however, potential sources of reporting bias—such as incomplete follow-up or missing imaging data—were qualitatively considered during synthesis.

Due to the heterogeneity of reported cases and the descriptive study designs, a meta-analysis could not be conducted. Instead, the findings were synthesized narratively, with descriptive statistics used where applicable, and results illustrated through tables and figures to highlight recurrent clinical patterns. The synthesis aimed to summarize demographic characteristics, diagnostic approaches, management strategies, and outcomes, while qualitatively addressing potential reporting biases and incomplete data. The study selection process is presented in the PRISMA flow diagram (Figure 1), and the review was conducted in accordance with the PRISMA 2020 guidelines, with the completed checklist included as Appendix A.

## 3. Results

From the 21 publications reviewed, a total of 24 individual cases of giant left atrium were identified. Whenever percentages are reported, the corresponding denominators are provided for clarity (e.g., “12/24 cases = 50%”). Measures of dispersion such as standard deviation or range were included when extractable from source reports; however, most values derived from single-case data, precluding calculation of uniform statistical measures. The mean age at presentation was 58.5 years, with a median of 60 years and a wide range between 22 and 89 years [17,18]. Most patients were in late middle age or elderly, but isolated reports also described younger individuals, reflecting the chronic progression of the condition. A female predominance was noted, with 15 women [5,18,19,20,21,22,23,24,25,26,27,28,29,30] and 8 men reported [17,24,31,32,33,34,35], while in one case sex was not specified [4]. This demographic distribution suggests that women are more frequently represented among the published cases.

Analysis of the temporal distribution of published cases showed variability in reporting across the last decade, with a modest clustering in certain years. When stratified by sex, female patients consistently outnumbered males. The grouped bar chart illustrates that, although cases were reported intermittently, the female-to-male ratio remained relatively stable over time (as shown in Figure 2). These findings emphasize both the rarity of giant left atrium and its demographic pattern, suggesting that women are more frequently affected in published reports.

These demographic findings reinforce the concept that giant left atrium is primarily encountered in older patients with a long-standing history of mitral valve pathology. The female predominance may reflect both the higher burden of rheumatic mitral valve disease among women and a potential sex-related susceptibility to atrial remodeling.

In the analyzed cases, dyspnea emerged as the leading symptom, present in the majority of patients and reflecting both elevated left atrial pressure and concomitant pulmonary hypertension. Palpitations were frequently reported and were most often related to atrial fibrillation, which affected nearly half of the cohort [17,24,31]. Signs of right-sided congestion, such as peripheral edema, were also observed, indicating the chronic hemodynamic burden of the markedly dilated chamber [18,20,26].

Compressive symptoms, although less frequent, carried particular diagnostic value. Dysphagia, stridor, and hoarseness were reported in a minority of patients, but they represented clear markers of extreme atrial enlargement with direct mechanical impact on adjacent structures [24,29]. Figure 3 illustrates the distribution of compressive and cardiorespiratory symptoms reported in the reviewed cases. These atypical presentations are of special interest, as they may lead to delayed recognition if not correlated with cardiac imaging [4,27].

The overall distribution, summarized in the donut chart, demonstrates the predominance of cardiorespiratory complaints, while highlighting the importance of vigilance for rarer extracardiac manifestations that can decisively influence clinical management.

Echocardiography represented the primary diagnostic tool in the majority of cases, being employed in two-thirds of the cohort [4,17,18,19,20,21,23,24,28,29,31,32,34,35]. Its central role reflects not only the widespread availability of the technique, but also its capacity to provide precise measurements of atrial dimensions, to characterize valvular pathology, and to assess hemodynamic consequences. Chest radiography was reported in a minority of patients and generally served as an initial screening modality, demonstrating gross cardiomegaly or displacement of adjacent structures. In a smaller subset, computed tomography was used as the first-line investigation, particularly in cases where compressive symptoms prompted evaluation of the mediastinum [22,24,26,27,30].

Complementary imaging was frequently required. Computed tomography was the most common adjunct, performed either in isolation or in combination with echocardiography, and offered superior delineation of the anatomical relationships between the enlarged atrium and neighboring organs such as the esophagus and bronchi. Cardiac magnetic resonance imaging was applied less frequently but provided incremental value in selected patients, especially when detailed volumetric quantification or tissue characterization was sought [4,5,19,29,33,35]. Advanced echocardiographic techniques, including speckle-tracking analysis, were reported only sporadically. Taken together, these data confirm the primacy of echocardiography in establishing the diagnosis, while cross-sectional imaging plays a crucial complementary role in documenting extracardiac compression and refining preoperative planning.

Among the 24 cases identified, the vast majority involved left atrium, consistent with existing literature. However, two patients presented with extreme right atrial dilatation, underscoring that this phenomenon, although exceptional, is not confined to left atrium. Clinical manifestations and complications were broadly similar, with arrhythmias and hemodynamic compromise being predominant, but compressive symptoms were less consistently reported in right-sided cases. These observations highlight the need to consider giant atrium as a spectrum that may affect either chamber, though left atrial involvement remains far more frequent.

Atrial fibrillation was the most frequent arrhythmic manifestation, documented in 12 patients, accounting for 50% of the cohort. This high prevalence confirms the strong arrhythmogenic potential of extreme atrial dilatation and its role as both a clinical marker and a prognostic factor [4,18,20,21,22,24,27,29,30,31,32,34,36]. Other arrhythmias were reported only sporadically and did not demonstrate a consistent pattern (as shown in Figure 4).

Associated conditions were heterogeneous. Hepatomegaly was described in three cases, reflecting chronic venous congestion, while systemic arterial hypertension was present in two patients, one of whom also had cirrhosis [18,24]. Less common associations included Hurler–Scheie syndrome [19], colon cancer [33], pneumonia [5], pleural effusion [26], and amaurosis [27]. Although individually rare, these comorbidities illustrate the diverse systemic contexts in which giant atrium may be encountered. Overall, the coexistence of atrial fibrillation and extracardiac comorbidities highlights the multifactorial burden carried by these patients and underscores the complexity of clinical management.

A small number of patients had a history of prior cardiac surgery before the diagnosis of giant atrium was established. Documented procedures included mitral valve replacement, excision of left atrial myxoma, and, in one instance, cardiac transplantation. In most cases, atrial enlargement persisted or progressed despite surgery, underscoring the chronic and progressive nature of atrial remodeling once established.

Intracavitary thrombus was identified in seven patients [4,18,19,20,22,34,35]. Among patients with documented thrombus, 71.4% (*n* = 5) had concomitant atrial fibrillation, reinforcing its role as the main prothrombotic driver in giant atrial chambers, supporting the concept that extreme atrial dilatation combined with electrical remodeling creates a prothrombotic substrate. Four patients (57.1%) with thrombus also had a history of prior cardiac surgery, yet no specific operative technique emerged as a consistent antecedent. Taken together, these findings suggest that atrial fibrillation represents the predominant risk factor for thrombus formation in giant atrium, while surgical history appears to play only a secondary role. “Early mortality” was defined as death occurring within 30 days postoperatively, and “thromboembolic events” included clinically confirmed stroke, transient ischemic attack, or systemic embolism. Follow-up duration was highly variable among reports, ranging from immediate postoperative observation to several years, limiting direct comparison between cases.

Echocardiographic data were available in a subset of cases. The mean anteroposterior atrial diameter was 13.4 cm, with extreme enlargement up to 20.6 cm (Figure 5). In four patients the dilatation was biatrial, while in most cases the enlargement was confined to left atrium [20,23,24]. Left ventricular systolic function was variably reported: although individual measurements suggested preserved or only mildly impaired ejection fraction in the majority, inconsistencies in reporting limited pooled analysis. Pulmonary artery hypertension was documented in 11 cases, most frequently graded as severe [18,19,20,24,25,26,27,30].

Therapeutic strategies varied widely across the reported cases. Most patients (*n* = 17) received conservative management consisting of anticoagulation, diuretic therapy, and rate control. Although this approach prevented further deterioration in several instances, atrial fibrillation and thromboembolic risk persisted, underscoring the limitations of medical therapy in this population.

Among the reviewed cases, atrial appendage aneurysm was a recurrent finding, either explicitly reported or documented through measurement of appendage dimensions [4,5,17,19,21,22,23,31,33]. While several patients were managed conservatively despite marked aneurysmal dilatation, others underwent surgical treatment, most frequently resection or closure of the aneurysm as part of a more complex procedure including atrial reduction or valve surgery. The choice of technique appeared to depend on concomitant pathology and surgical indication, rather than on appendage size alone. These observations highlight both the frequent involvement of the appendage in the context of giant atrium and the lack of a standardized operative strategy for this anatomical substrate.

Surgical management was reported in six patients and encompassed a heterogeneous spectrum of procedures. Atrial reduction was performed in three cases [24], usually in association with mitral valve surgery, aiming to decrease chamber volume and alleviate compressive symptoms. Reconstructive approaches were described in another three patients, including techniques addressing the atrial wall or auricular aneurysm. In several instances, closure or resection of the appendage was performed concomitantly, reflecting its frequent aneurysmal involvement and potential thromboembolic risk.

Postoperative outcomes were available for six surgically treated patients. In this subgroup, five procedures were not followed by early complications, while one patient developed pneumonia and mesenteric ischemia. The distribution of published cases is documented in Appendix B, Table A1. Survival ranged from 6 months to more than 10 years, although three patients eventually died, two in late follow-up and one within 202 days of surgery. In contrast, among the 17 patients managed conservatively, complication data were largely unavailable; however, isolated reports described heart failure, arrhythmia, and multiorgan dysfunction. Early mortality was noted in two conservatively treated patients, both within the first two weeks of follow-up [28,35]. A summary of treatment modalities and reported outcomes is presented in Table 1 and Table 2, illustrating the relationship between management strategy and major clinical endpoints.

These findings highlight the clinical heterogeneity of giant atrial dilatation, the diagnostic primacy of echocardiography, and the need for tailored treatment approaches. The reviewed cases demonstrated substantial variability in etiology, symptom presentation, and management strategies, reflecting the absence of standardized definitions and therapeutic consensus. Despite these differences, echocardiography consistently emerged as the principal diagnostic modality, while cross-sectional imaging provided valuable complementary information in patients with suspected extracardiac compression.

## 4. Case Presentation

A 51-year-old male patient with a history of severe left atrial enlargement and secondary dilated cardiomyopathy related to valvular disease underwent double-valve replacement surgery in 2019 at our institution, receiving a mechanical mitral prosthesis and a biological tricuspid prosthesis. At that time, surgical intervention had been delayed for personal reasons until left atrium was already markedly enlarged, although left ventricular systolic function was preserved. Following surgery, the patient did not attend regular follow-up.

In 2025, he presented to our clinic with progressive dyspnea, dysphagia, and severe dysphonia as the main complaints. Clinical evaluation revealed decompensated heart failure, chronic liver disease with cirrhosis and splenomegaly, and chronic kidney disease. He was in permanent atrial fibrillation with a controlled ventricular rate.

Transthoracic echocardiography showed a severely dilated left atrium, measuring 10.3 cm in diameter with an estimated volume of approximately 2000 mL (as shown in Figure 6). The left ventricle was dilated and hypertrophied, globally hypokinetic, with a reduced ejection fraction of 35–40%. The mitral mechanical prosthesis and tricuspid bioprosthesis were both normally functioning, with only mild intraprosthetic tricuspid regurgitation. Degenerative aortic regurgitation of moderate severity was also present. The right ventricle was dilated with systolic dysfunction and reduced TAPSE, while pulmonary artery pressures were severely elevated but could not be precisely quantified.

A preoperative non-contrast chest CT confirmed massive cardiomegaly, predominantly due to left atrial enlargement measuring 11 × 19.2 cm, increased compared to a previous scan in December 2022 (9.2 × 15.5 cm). The atrium displaced the esophagus to the right, elevated the carina, and compressed the left lower lobe bronchus, leading to atelectasis. Additional findings included a pericardial effusion of up to 1.2 cm, a moderate left pleural effusion, and impaired ventilation of the left lower lobe. The examination was performed without intravenous contrast administration owing to coexisting renal impairment (see Figure 7).

Abdominal ultrasound revealed hepatomegaly with advanced fibrosis, portal vein dilatation, splenomegaly, and moderate ascites, consistent with congestive hepatopathy and possible cardiac cirrhosis.

During hospitalization, the patient experienced repeated decompensations affecting the cardiac, renal, hepatic, and respiratory systems, with bilateral pleural effusions requiring multiple thoracenteses and suspected bilateral pneumonia. He required prolonged inotropic support.

After multidisciplinary evaluation, the patient and his family consented to high-risk surgical intervention. A left atrial reduction was performed using linear endoplication reinforced with pledgeted mattress sutures and prolene sutures (as shown in Figure 8 and Figure 9).

Postoperatively, the patient remained in critical condition, intubated, and dependent on maximal inotropic and vasopressor support. Early echocardiography showed compression of the right-sided cardiac chambers with features of impending collapse. Preoperative laboratory parameters, including coagulation profile, were within acceptable limits and consistent with a compensated clinical status, while postoperative deterioration reflected the onset of acute hemodynamic compromise. The associated coagulation disorders, secondary to hepatic dysfunction, further amplified the hemodynamic instability and contributed to the unfavorable evolution of the physiopathological cascade. Emergency re-exploration revealed hemopericardium with tamponade, which was evacuated with surgical drainage. Perioperative anticoagulation was adjusted and monitored according to institutional protocols, but the patient’s condition remained unstable despite maximal supportive therapy. He subsequently suffered an irreversible cardiac arrest due to electromechanical dissociation, unresponsive to advanced resuscitation.

Taken together, these findings highlight the diverse clinical spectrum of giant atrial dilatation, the central role of echocardiography in diagnosis, and the continued need for individualized therapeutic approaches in the absence of standardized guidelines.

## 5. Discussion

The present synthesis emphasizes that giant atrial dilatation, although uncommon, is a condition with significant clinical and prognostic implications. Historically described in the context of rheumatic mitral valve disease, it is now recognized across a broader spectrum of etiologies, including congenital malformations, degenerative valvular lesions, and idiopathic enlargement [37]. This demographic pattern aligns with previous reports suggesting sex-specific susceptibility to chronic atrial remodeling [9,10,37].

Clinically, dyspnea and palpitations constitute the predominant symptoms, reflecting both hemodynamic congestion and the high incidence of atrial fibrillation. Extracardiac manifestations such as dysphagia, stridor, or hoarseness are less common but remain clinically significant. These atypical presentations are particularly relevant for gastroenterologists and pulmonologists, as failure to consider cardiac enlargement in the differential diagnosis may delay recognition [1,4,32,38,39,40]. Our institutional case, marked by severe dysphagia due to esophageal compression, exemplifies the diagnostic and therapeutic challenges posed by such atypical manifestations.

Atrial fibrillation was documented in 50% of patients, closely associated with thrombus formation in one-third of cases. This correlation highlights the dual substrate of stasis and electrical remodeling inherent to giant atria, which fosters both arrhythmia and thrombogenesis. Pulmonary hypertension, severe in many cases, further compounds the hemodynamic burden. Collectively, these features place patients at high risk of stroke, right-sided heart failure, and systemic organ dysfunction [19,21,41,42,43,44].

Therapeutic strategies remain heterogeneous and controversial. Conservative treatment, consisting of anticoagulation, diuretics, and rate control, was the most frequently employed approach, yet it did not reliably prevent progression or thromboembolic events [41,45,46,47]. Surgical interventions—including atrial reduction, aneurysm resection, and valve procedures—were reported in a minority of cases [46,48]. Although surgery offers the potential to relieve compressive symptoms and reduce thromboembolic risk, postoperative outcomes have shown marked variability. In our case, the patient did not survive despite timely surgical reduction. This outcome underscores both the technical challenges of atrial reduction procedures and the significant impact of advanced comorbidities on prognosis [13,15,16,46,48].

The decision to operate should therefore be individualized, taking into account symptom severity, hemodynamic compromise, arrhythmic burden, and extracardiac manifestations. Multidisciplinary evaluation, ideally involving cardiologists, surgeons, and imaging specialists, is essential. Equally important is early recognition; delayed referral, as in our patient, limits the potential benefits of surgery and increases procedural risk [44,49].

To translate current evidence into practical clinical guidance, we propose a simplified diagnostic and management algorithm (Figure 10). This flowchart outlines the recommended pathway for evaluating patients with suspected giant atrial dilatation, beginning with echocardiographic confirmation and extending to etiologic assessment, risk stratification, and individualized treatment selection. Conservative management is appropriate for stable or high-risk patients without compressive symptoms, while surgery should be considered in those with compressive or embolic complications, concomitant valve disease, or progressive enlargement despite optimal medical therapy. The final step emphasizes structured follow-up and standardized data recording—such as atrial dimensions, rhythm status, treatment type, and outcomes—to facilitate future registry-based research and improve comparability across studies.

Our review has several strengths, including a structured search strategy, defined inclusion criteria, and systematic data extraction, which enhance transparency and reproducibility. However, several limitations must be acknowledged. The analysis was restricted to a single database (PubMed) and to open-access English-language publications, which may have introduced selection and language bias. The available evidence consists mainly of individual case reports and small series with heterogeneous design, incomplete follow-up, and variable reporting quality, precluding quantitative synthesis or meta-analysis. Definitions of “giant” atrium remain inconsistent—particularly for right atrium and atrial appendage aneurysms—making cross-study comparisons difficult. Publication bias is also possible, as unusual or surgically treated cases are more likely to be reported.

Despite these constraints, our synthesis provides a comprehensive and clinically relevant overview of current evidence, highlighting diagnostic patterns, management dilemmas, and practical implications derived from both literature and institutional experience.

Extreme atrial dilatation should be regarded not merely as an anatomical curiosity, but as a clinically significant condition with important implications for morbidity and mortality. Our synthesis of 24 published cases, supplemented by our institutional experience, reveals a consistent demographic pattern—predominantly middle-aged to elderly women with underlying rheumatic mitral valve disease. Arrhythmias, particularly atrial fibrillation, represented the most prevalent complication, documented in approximately 50% of cases, highlighting the need for vigilant rhythm monitoring and systematic anticoagulation. Extracardiac compression, though less frequent, emerged as a critical clinical milestone, emphasizing the role of multimodality imaging in recognizing atypical presentations.

Future research should prioritize establishing standardized diagnostic criteria, developing consensus-driven surgical indications, and prospectively evaluating outcomes in patients managed conservatively or surgically. Advances in imaging and surgical techniques hold promise to improve risk stratification and broaden treatment options for this rare but clinically significant pathology [16,50,51]. In parallel, translational studies exploring the molecular and biomechanical remodeling of the atrial wall—particularly alterations in biomechanical signaling pathways and extracellular matrix composition (e.g., AT1R, TGF-β, Collagen I/III)—may provide deeper insight into disease progression and identify potential antifibrotic therapeutic targets [1,44].

## 6. Conclusions

Extreme atrial dilatation should be regarded not merely as an anatomical curiosity, but as a clinically significant condition with important implications for morbidity and mortality. Our synthesis of 24 published cases, supplemented by our institutional experience, reveals a consistent demographic pattern—predominantly middle-aged to elderly women with underlying rheumatic mitral valve disease. A recurrent constellation of complications was identified, most notably atrial fibrillation, thromboembolic events, and pulmonary hypertension.

Although less frequently observed, extracardiac compression—manifesting as dysphagia, stridor, or airway obstruction—represents a critical diagnostic clue often prompting advanced imaging and intervention

Arrhythmias, particularly atrial fibrillation, represented the most prevalent complication, documented in approximately 50% of cases. Its strong association with intracavitary thrombus formation underscores the interplay of structural and electrical remodeling inherent to giant atria. Additionally, pulmonary hypertension—frequently severe—further compounds the hemodynamic burden, contributing to right-sided heart failure and systemic congestion. These findings highlight the importance of vigilant rhythm monitoring and appropriate anticoagulation in all patients with extreme atrial enlargement.

Therapeutic strategies remain heterogeneous. Conservative management is most commonly employed and may offer symptomatic relief; however, it does not reliably prevent disease progression or reduce long-term thromboembolic risk. Surgical interventions—although potentially effective in relieving compressive symptoms and reducing atrial volume—are associated with substantial perioperative risk, particularly in patients with advanced comorbidities or multi-organ dysfunction. Our institutional case underscores the prognostic impact of delayed referral, which may limit the benefits of surgery despite technical success.

In clinical practice, echocardiography should remain the first-line diagnostic tool, complemented by cross-sectional imaging when compressive symptoms are present.

Management decisions should be guided by an integrated assessment of arrhythmic burden, thromboembolic risk, and the presence of extracardiac compression. Given the high prevalence of intracavitary thrombi, anticoagulation is advisable for all patients, even those maintaining sinus rhythm. Conservative therapy may be appropriate for stable, asymptomatic patients, particularly when surgical risk is prohibitive. However, surgical intervention should be strongly considered in cases of persistent compressive symptoms, when there are recurrent embolic events despite adequate anticoagulation, or when concomitant valve surgery is indicated. Timely intervention before irreversible end-organ damage remains critical for improving outcomes

Future research should aim to establish standardized diagnostic criteria and prospective registries to refine risk stratification and optimize long-term management. Giant atrial pathology thus constitutes a rare but clinically significant condition at the interface of valvular, arrhythmic, and extracardiac diseases, where early recognition and risk-adapted treatment are pivotal for improving patient prognosis.

## Figures and Tables

**Figure 1 jcm-14-07832-f001:**
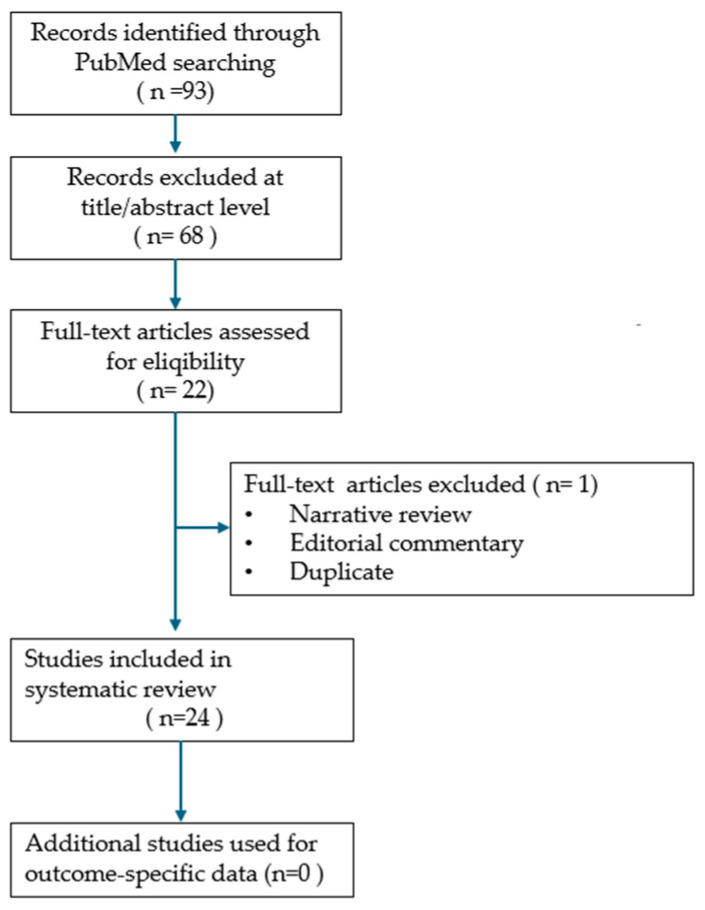
PRISMA flow diagram of study selection process.

**Figure 2 jcm-14-07832-f002:**
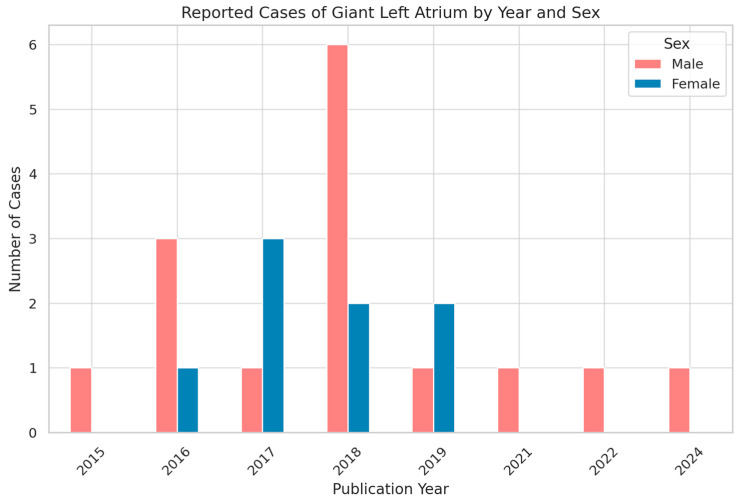
Temporal and Sex Distribution of Reported Giant Left Atrium Cases (2015–2024).

**Figure 3 jcm-14-07832-f003:**
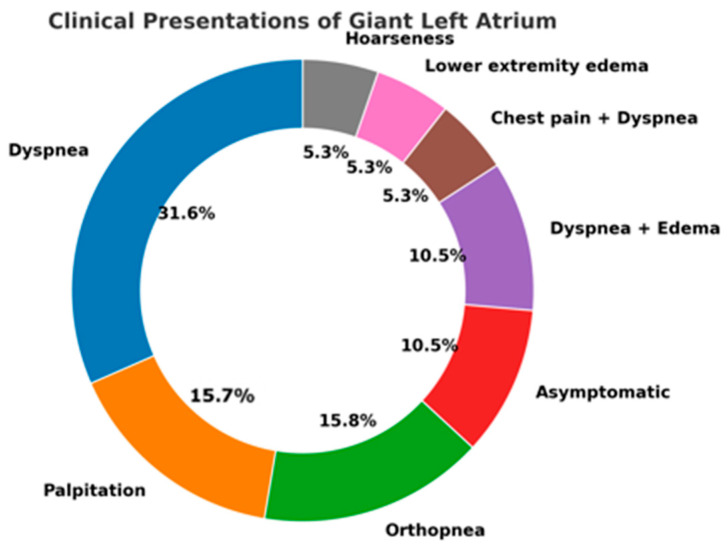
Spectrum of Clinical Presentations in Reported Cases of Giant Left Atrium.

**Figure 4 jcm-14-07832-f004:**
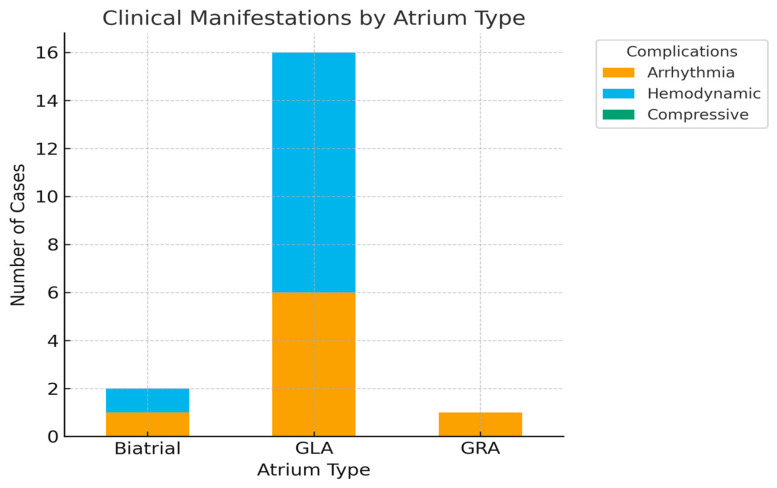
Distribution of Clinical Complications by Atrium Type.

**Figure 5 jcm-14-07832-f005:**
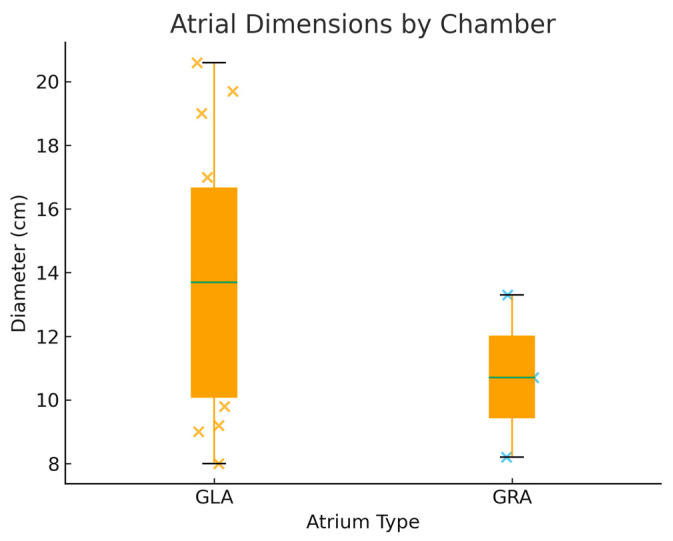
Boxplot of Reported Atrial Diameters: Giant Left versus Giant Right Atrium.

**Figure 6 jcm-14-07832-f006:**
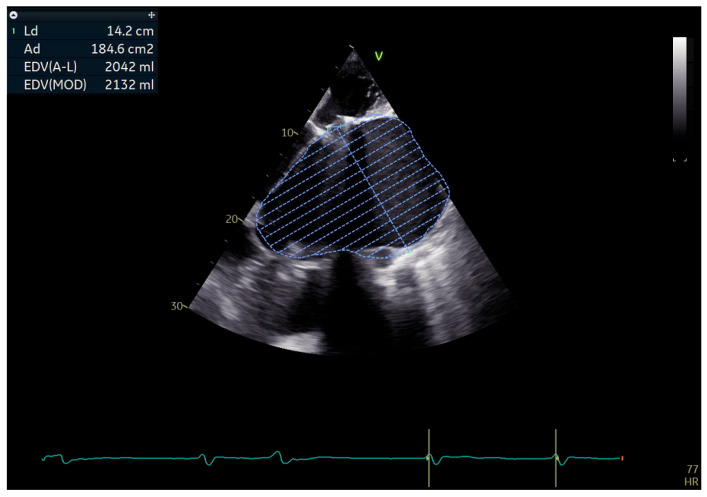
Apical echocardiographic view showing a massively dilated left atrium with traced endocardial border and calculated volume > 2000 mL.

**Figure 7 jcm-14-07832-f007:**
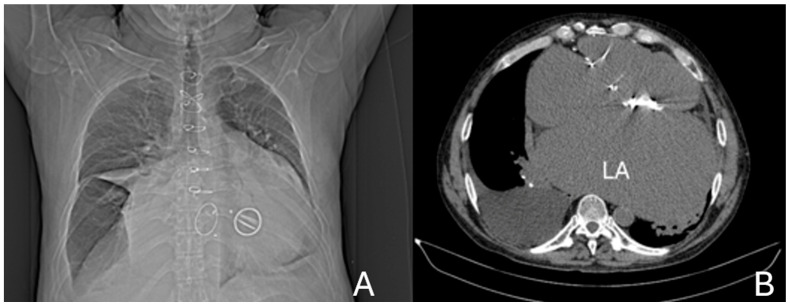
(**A**) Anterior CT scout view, demonstrating marked cardiomegaly, the prosthetic mitral valve, and the skeletal frame of a bioprosthetic tricuspid valve. (**B**) Axial non-contrast CT image at the mid-thoracic level showing severe left atrial (LA) enlargement with a prosthetic mitral valve in situ.

**Figure 8 jcm-14-07832-f008:**
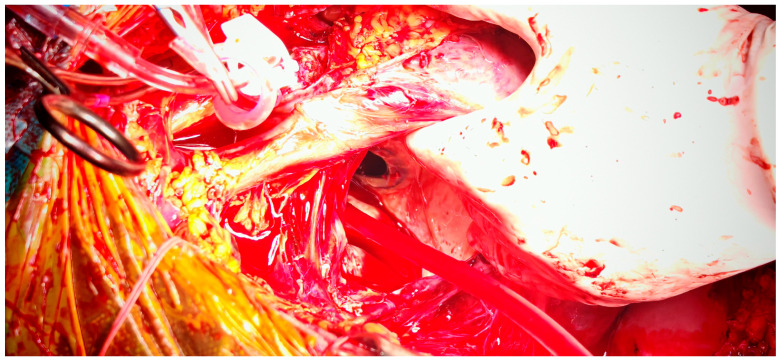
Intraoperative view of left atrium after atriotomy, with exposure of the prosthetic mitral valve.

**Figure 9 jcm-14-07832-f009:**
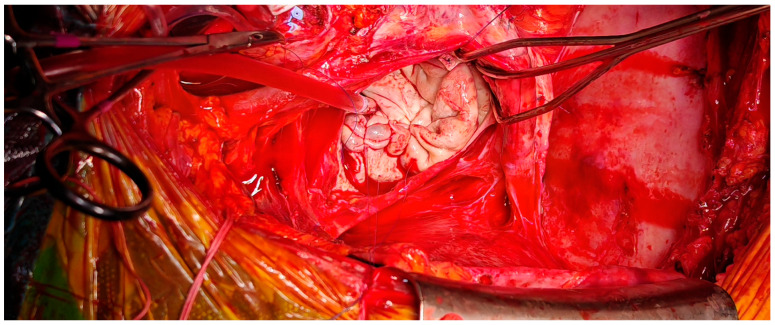
Intraoperative aspect of left atrium following plication, demonstrating reduction in atrial cavity size.

**Figure 10 jcm-14-07832-f010:**
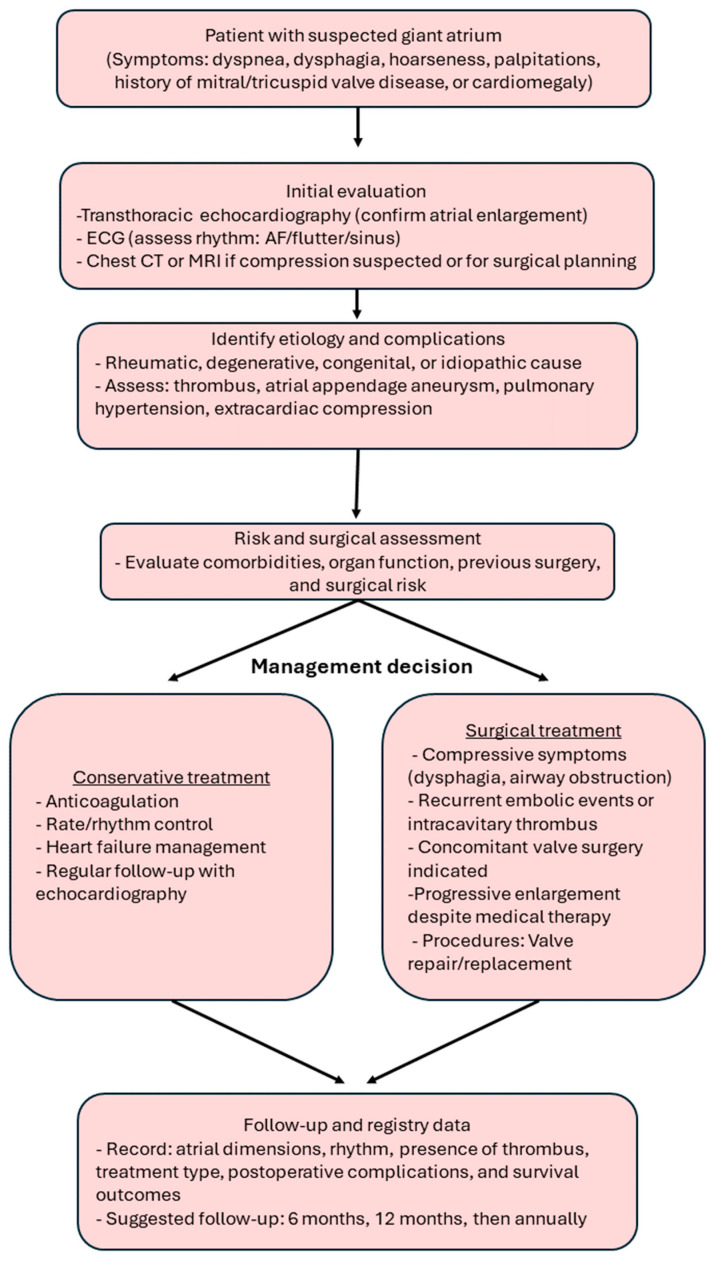
Diagnostic and management algorithm for giant atrial dilatation. The flowchart summarizes clinical evaluation, risk assessment, treatment decision-making, and key follow-up parameters for future registry data collection.

**Table 1 jcm-14-07832-t001:** Reported mortality by management type in published cases of giant atrial dilatation.

*Management Type*	*No. of Patients*	*Early Mortality **	*Late Mortality*
*Surgical* (atrial reduction ± valve surgery)	6	1 (16.7%)	2 (33.3%)
**Conservative** (medical therapy, anticoagulation, diuretics)	17	2 (11.8%)	Not reported

* Early mortality defined as death within 30 days postoperatively.

**Table 2 jcm-14-07832-t002:** Reported complications and follow-up duration by management type.

*Management Type*	*Major Complications*	*Follow-Up Duration*
*Surgical*	Pneumonia, mesenteric ischemia	6 mo–10 yr
*Conservative*	Heart failure, arrhythmia, multiorgan dysfunction (isolated reports)	Weeks–years

## Data Availability

The original contributions presented in this study are included in the article. Further inquiries can be directed to the corresponding authors.

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
