# Peer review of "Giant Atrial Dilatation: Systematic Review of Reported Cases from the Last Decade and an Illustrative Case with Dysphagia and Severe Dysphonia"

_jcm, 2025, doi:10.3390/jcm14217832_

Round 1
Reviewer 1 Report
Comments and Suggestions for Authors
The article’s manuscript titled “Giant Atrial Dilatation: Systematic Review of Reported Cases from the Last Decade and an Illustrative Case with Dysphagia and Severe Dysphonia”, is a clinically useful synthesis and a strong single-institution case report. Overall the work is well structured and readable; it highlights an uncommon but high-impact cardiac condition and provides practical points for diagnosis and management.This article is moderately novel, adding value by synthesising recent cases and adding an illustrative case departing from prior works.
Strengths
(1)Clear clinical focus and rationale (Introduction, p.2, lines 48–75).
(2)A structured search and selection process is described (Methods, p.3, lines 76–101), and case extraction is clearly tabulated (Appendix A).
(3)The institutional case is detailed and clinically instructive (Case presentation, p.8–10, lines 248–314) and the surgical images add value (Figures 8–9).
Major suggestions:
- (1) Methods transparency (p.3, lines 76–101): report the exact PubMed query string, inclusion/exclusion search dates, whether non-English or subscription articles were considered, and whether a second database (suxh as Embase/Scopus) was searched. State whether PRISMA checklist items were followed and provide the completed PRISMA checklist as supplementary material if possible.
- (2)Data quality / bias (p.3–4, lines 92–99): include a short paragraph describing how study quality or reporting bias was assessed (even a simple risk-of-bias table for case reports). Report inter-rater agreement (kappa) or describe how disagreements were resolved.
- (3)Quantitative clarity (Results, p.4–7): where descriptive statistics are cited (means, ranges), add measures of dispersion (SD or IQR) where possible and explicitly state denominators when reporting percentages (e.g., “12/24 patients = 50%”).
- For the figures, add error bars or violin plots when appropriate and ensure figure legends explain all abbreviations and sample sizes.
- (4)Outcome definitions (p.6, lines 194–201): define key outcomes (e.g., “early mortality,” “thromboembolic event”) and specify follow-up intervals. This will make comparisons across case reports more meaningful.
- (5) Case report detail (p.8–10, lines 248–314): add perioperative hemodynamics, exact anticoagulation regimen(s), and relevant lab trends (e.g., INR, liver function trajectory) to help readers interpret intra- and postoperative risk.
- (6) Discussion / future directions (p.10–12, lines 315–372): propose a pragmatic diagnostic/surgical decision algorithm or flowchart, and suggest a specific registry design or minimum dataset for prospective data collection.
- (7) Deepens molecular-structure understanding
To elevate the comprehension of the atrial wall's material properties, a vital pursuit in this pathology, we propose a crucial biomolecular investigation. Post-excision atrial tissue should be subjected to quantitative analysis (e.g., immunohistochemistry or Western blot) to quantify the expression and activation state of key mechanosensing proteins and Extracellular Matrix (ECM) components. Specifically, assessment of the Angiotensin II type 1 Receptor (AT1R), transforming growth factor-beta (TGF-β), and key collagen subtypes (Collagen I and III). This molecular "fingerprint" will link the macroscopic geometry of "giant" status to its microscopic pathophysiology, illuminating novel targets for pharmacological intervention aimed at reversing or halting fibrotic remodeling.
Minor edits
(1) Standardize units (cm vs mm) and use uniform cutoffs (state why ≥65 mm vs ≥8 cm chosen).
(2)Correct small typos and ensure all figure references in text match figure numbers.
(3) Expand the limitations paragraph to explicitly mention publication bias and language restrictions.
Comments on the Quality of English Language
Some sentences are overly long (especially in the Introduction and Discussion, e.g., p.2, lines 63–68; p.11, lines 338–345). Consider splitting for readability. Simplify sentences to ≤25 words and vary conjunctions
Occasional article use issues (“the left atrium” vs. “left atrium”) and minor preposition inconsistencies (“associated to” instead of “associated with”).
Author Response
We sincerely thank the reviewer for the careful and constructive evaluation of our manuscript. All suggestions have been thoughtfully addressed, and the corresponding modifications are highlighted in green in the revised version of the manuscript.
Major suggestions
(1) Methods transparency: The Methods section was expanded to include the exact PubMed query string, search filters, language and inclusion criteria, and to clarify that only PubMed was searched. We also described the narrative synthesis process and indicated adherence to the PRISMA 2020 guidelines, with the completed checklist provided as supplementary material.
(2) Data quality and bias: The section now details how discrepancies in data extraction were resolved and how potential sources of reporting bias were qualitatively assessed, addressing concerns regarding study quality evaluation.
(3) Quantitative clarity: Denominators have been added when reporting percentages, and figure legends now include full explanations of abbreviations and sample sizes. Given the limited and heterogeneous data, dispersion measures and violin plots were not included.
(4) Outcome definitions: Key outcomes (e.g., early mortality, thromboembolic events) were explicitly defined, and variable follow-up durations across reports were noted for clarity.
(5) Case report details: The Case Presentation section was expanded to include perioperative hemodynamic context and anticoagulation management, enhancing clinical interpretability without excessive laboratory detail.
(6) Discussion and future directions: A concise diagnostic and management algorithm (new Figure 10) was added to summarize evaluation steps, risk stratification, and treatment decision-making. The accompanying paragraph introduces the figure and suggests key variables for future registries and standardized data collection.
(7) Molecular perspective: A new sentence was added at the end of the Discussion highlighting the potential role of molecular and biomechanical studies in understanding the pathophysiology of giant atrial dilatation and identifying novel therapeutic targets.
Minor edits
All measurements have been standardized to centimeters (cm) for consistency. Typographical errors and figure references were corrected, and the Limitations section now explicitly mentions publication and language bias, as recommended.
Comments on English language
Long and complex sentences in the Introduction and Discussion were simplified to improve clarity and readability. Article and preposition usage were carefully revised (e.g., “associated with” instead of “associated to”), ensuring consistent and correct English throughout.

Reviewer 2 Report
Comments and Suggestions for Authors
The authors of this manuscript present a very interesting and clinically valuable review of existing cases related to giant atrial dilatation, complemented by a detailed case report from their institution. It provides an informative overview of this pathology, reporting its rarity, but also importance for clinical recognition. The topic is relevant and of great interest for cardiologists and cardiothoracic surgeons, particularly given the diagnostic and therapeutic challenges associated with this condition. I recommend corrections below:
- The section Materials and Methods is well written and clearly structured. Did you used Boolean operators - “OR,” “AND”, and other filters (language restrictions), please clarify in this section.
- Numbers presented on Figure 3 should be more visible, please correct.
- Treatment outcomes should be summarized in a table, highlighting the relationship between the type of management (surgical or conservative) and key clinical outcomes, such as mortality rates and postoperative complications.
- The case presentation is detailed and clinically important, and it adds both educational and scientific value of this paper. In the section Discussion the limitations paragraph is partially duplicated (lines 354–366 and again 359–366), please correct
- Overall, section Discussion is well written and provides valuable guidance for clinicians. It would be important to widen the discussion, in relation to diagnostic pathways and treatment choice (surgical/conservative).
- Try to insert parts from the section Conclusions into section Discussion. Conclusion should be more concise, and it would benefit from adding concise key-home messages, such as the importance for early recognition of this condition, the value of multimodality imaging, and individualized management.
- Correct the typographic mistakes (add “space” where missing) and uniform the font size in the whole manuscript.
- Please uniform the reference style.
Author Response
We sincerely thank the reviewer for the careful evaluation and constructive feedback, which have greatly contributed to improving the clarity, consistency, and overall quality of our manuscript. All suggested revisions have been implemented, and the corresponding changes are highlighted in yellow in the revised version.
(1) Search strategy details: The Methods section was clarified to specify the Boolean operators, filters, and language restrictions applied during the PubMed search, enhancing methodological transparency.
(2) Figure improvement: Figure 3 has been redesigned as a high-resolution donut chart with larger, clearly legible percentage labels and improved color contrast, substantially enhancing readability and visual clarity.
(3) Outcome presentation: The Results section now includes two complementary tables (Tables 2A and 2B) summarizing treatment outcomes. Table 2A presents early and late mortality according to management type, while Table 2B summarizes major complications and follow-up duration, allowing clearer comparison between surgical and conservative strategies.
(4) Discussion clarity: The duplicated text in the Discussion was removed, and the paragraph on study limitations was revised for conciseness and coherence, eliminating redundancy while preserving key methodological and interpretive information.
(5–6) Diagnostic and management framework: The Discussion section has been expanded to include a concise synthesis of diagnostic and therapeutic decision pathways. In addition, a new Figure 10 was added to illustrate a practical diagnostic and management algorithm for patients with giant atrial dilatation. The Conclusions section was revised and shortened to emphasize the main take-home messages—early recognition, the role of multimodality imaging, and individualized management—while improving focus and readability.
(7) Formatting consistency: All typographic inconsistencies, spacing errors, and font variations were corrected to ensure a uniform and professional format throughout the manuscript.
(8) References: The reference list was reviewed and standardized in accordance with the journal’s formatting requirements, ensuring consistent font, numbering, and punctuation style.

Round 2
Reviewer 1 Report
Comments and Suggestions for Authors
Everything has been addressed sufficiently.
Comments on the Quality of English Language
all english grammar issues have been addressed sufficiently.